# Fabrication of Mn-Doped SrTiO_3_/Carbon Fiber with Oxygen Vacancy for Enhanced Photocatalytic Hydrogen Evolution

**DOI:** 10.3390/ma15134723

**Published:** 2022-07-05

**Authors:** Qi Hu, Jiantao Niu, Ke-Qin Zhang, Mu Yao

**Affiliations:** 1National Engineering Laboratory for Modern Silk, College of Textile and Clothing Engineering, Soochow University, Suzhou 215123, China; huqi@usts.edu.cn; 2School of Social Development and Public Administration, Suzhou University of Science and Technology, Suzhou 215009, China; 3School of Textile and Clothing and Arts and Media, Suzhou Institute of Trade and Commerce, Suzhou 215009, China; tjpunjt@126.com; 4School of Textile Science and Engineering, Xi’an Polytechnic University, Xi’an 710048, China; yaomu1930@163.com

**Keywords:** SrTiO_3_, hydrogen evolution, carbon fiber, oxygen vacancy

## Abstract

With carbon fiber, it is difficult to load semiconductor photocatalysts and easy to shed off thanks to its smooth surface and few active groups, which has always been a problem in the synthesis of photocatalysts. In the study, SrTiO_3_ nanoparticles were loaded onto the Tencel fibers using the solvothermal method, and then the Tencel fibers were carbonized at a high temperature under the condition of inert gas to form carbon fibers, thus SrTiO_3_@CF photocatalytic composite materials with solid core shell structure were prepared. Meanwhile, Mn ions were added into the SrTiO_3_ precursor reagent in the solvothermal experiment to prepare Mn-doped Mn-SrTiO_3_@CF photocatalytic composite material. XPS and EPR tests showed that the prepared Mn-SrTiO_3_@CF photocatalytic composite was rich in oxygen vacancies. The existence of these oxygen vacancies formed oxygen defect states (VOs) below the conduction band, which constituted the capture center of photogenerated electrons and significantly improved the photocatalytic activity. The photocatalytic hydrogen experimental results showed that the photocatalytic hydrogen production capacity of Mn-SrTiO_3_@CF composite material with 5% Mn-doped was six times that of the SrTiO_3_@CF material, and the doping of Mn ions not only promoted the red shift of the light absorption boundary and the extension to visible light, but also improved the separation and migration efficiency of photocarriers. In the paper, the preparation method solves the difficulty of loading photocatalysts on CF and provides a new design method for the recycling of catalysts, and we improve the hydrogen production performance of photocatalysts by Mn-doped modification and the introduction of oxygen vacancies, which provides a theoretical method for the practical application of hydrogen energy.

## 1. Introduction

As a sustainable green technology, semiconductor photocatalysis can transform solar energy, purify the environment, and produce renewable energy. At present, there are various kinds of semiconductor photocatalysts. Among them, SrTiO_3_ semiconductor material has been widely used in fields such as photocatalytic water splitting hydrogen production, virus inactivation, and pollutant treatment thanks to its characteristics of excellent chemical stability, non-toxicity, low cost, good photoelectric property, and environmental friendliness [1]. However, powder SrTiO_3_ photocatalytic material is easy to condense into blocks in use, which reduces the photocatalytic property of SrTiO_3_, and causes some problems such as difficulties in the recycling of catalyst, easy consumption, possible secondary pollution, and so on, limiting its practical application. Therefore, some researchers have begun to think about loading photocatalysts on more suitable carriers [2]. Carbon fiber (CF) has the advantages of a large specific surface area, high specific strength, high flexibility, and strong response to visible light, greatly improving the photocatalytic activity and stability of semiconductor photocatalysts; therefore, it is an ideal carrier for supporting granular semiconductor photocatalysts [3,4,5].

Similar to TiO_2_ photocatalysts, pure SrTiO_3_ has a large band gap (about 3.2 eV), resulting in its response to ultraviolet light accounting for a small amount of energy of sunlight, which severely limits the application of SrTiO_3_ in the field of visible light photocatalysis. Therefore, to improve the photocatalytic property of SrTiO_3_, modification strategies such as ion doping, co-catalyst loading, surface defect modification, and semiconductor heterojunction compounding have been proposed and further studied by researchers [1,6]. Ion doping is considered to be an effective method to improve the photocatalytic property of SrTiO_3_. Common metal ions doped include Pt, Ag, Au, Ru, Pd, Cu, Al, Mn, and so on [7,8,9,10,11,12,13,14]. Mn-doped SrTiO_3_ photocatalytic materials can reduce the band gap, while the preparation technology is mature and has been well developed and applied in the field of photocatalysis and widely used in the fields of photocatalytic degradation and photocatalytic water splitting hydrogen production [13,15,16]. Another effective method to improve the photocatalytic property of SrTiO_3_ is surface defect modification [17]. Oxygen vacancy is a research hotspot in defect engineering. The introduction of oxygen vacancy can solve the problems of low light absorption efficiency of photocatalysts, rapid recombination of photoelectrons and holes, and lack of surface active sites, and is proved to be an important and effective strategy to improve photocatalytic efficiency [18,19,20,21]. Therefore, the introduction of oxygen vacancies in the preparation of SrTiO_3_ photocatalytic material has attracted the attention of researchers.

How to combine the method of regulating the band gap by doping with the method of constructing surface defects by introducing oxygen vacancies is one kind of research approach to improve the photocatalytic property of SrTiO_3_. In addition, carbon fibers (CFs) can be loaded into SrTiO_3_ photocatalytic composite material as a photocatalyst carrier in consideration of difficulties in the recovery and recycling of photocatalysts. Previous studies on carbon fibers as a photocatalyst carrier have been reported, which basically found the following: carbon fibers were prepared first, and then the photocatalyst was loaded onto the carbon fibers. Such a method has problems including poor load stability and easy shedding of semiconductor materials [22,23,24]. How to prepare a highly stable SrTiO_3_/carbon fiber by firmly loading the catalyst onto carbon fibers is a research focus of this paper.

In this paper, the two methods of Mn doping and introduction of oxygen vacancies were combined. With Tencel fibers as the carrier, an Mn-doped SrTiO_3_ photocatalyst was loaded onto the surface of Tencel fibers by the solvothermal method, then the Tencel fibers were carbonized under the condition of argon, Mn-SrTiO_3_@CF with a core shell structure rich in oxygen vacancies was successfully prepared, and the photocatalytic hydrogen production mechanism was discussed.

## 2. Experiment Section

### 2.1. Preparation of Photocatalytic Material

Preparation of Mn-SrTiO_3_@Tencel Fiber Composite Fiber. The surface of Tencel fibers was coated with Mn-doped strontium titanate photocatalyst by the solvothermal method. Here, 15 mL of ethylene glycol and 19 mL of absolute ethyl alcohol were mixed and stirred well to form a mixed solvent. Then, 0.036 g of manganese acetate and 0.58 g of strontium acetate were added to the mixed solvent and stirred at room temperature for 30 min, and the above mixed solution was transferred to the PTFE reactor, wherein the molar ratio of manganese acetate to strontium acetate was 1:19. Then, 0.85 g of isopropyl titanate drop solution was added to the above mixed solution and it was stirred for one more hour [25]. After that, 0.8 g of Tencel fibers was selected, torn by hands, and then completely immersed into the solution in the PTFE reactors. The reactors were put into an oven for solvothermal reaction at 180 °C for 7 h. After the reaction finished and the temperature was reduced to room temperature, the Tencel fibers were removed, cleaned with absolute ethyl alcohol, and then dried in the oven at 60 °C for 24 h, thus obtaining the Mn-SrTiO_3_@Tencel fiber material.

Similarly, without manganese acetate reagent, SrTiO_3_@Tencel fiber composite fibers were prepared according to the above experimental method, and the pure SrTiO_3_ nanometer material was also prepared.

Preparation of Mn-SrTiO_3_@CF Composite Fiber. Under the condition of inert gas, the tencel fibers were carbonized at a high temperature to prepare carbon fibers [26], which transformed Mn-SrTiO_3_@Tencel fiber into Mn-SrTiO_3_@CF material, and the specific experimental steps were as follows. First, the Mn-SrTiO_3_@Tencel fiber was put into a crucible and then into a tube furnace, and the Tencel fibers were carbonized at 800 °C for 2 h under the condition of argon atmosphere while the heating rate was 5 °C/min, thus Mn-SrTiO_3_@CF composite material with a core shell structure was obtained.

Under the same process conditions, SrTiO_3_@CF was prepared.

### 2.2. Characterization Method

The crystalline structures of the samples were characterized by an X-ray diffractometer (XRD, Bruker D8 Disvover, Bruker, Karlsruhe, Germany) at a scanning speed of 0.2 °/min, and the morphologies of the as-prepared products were characterized by an electron microscope (FESEM Hitachi S-4800) and a transmission electron microscope (TEM FEI talos F200x G2). The binding energy was detected by X-ray photoelectron spectroscopy (XPS, Thermo Scientific K-Alpha, Thermo Fisher Scientific, Horsham, UK), and the defects of the samples were tested and characterized by an electronic paramagnetic spectrometer (EPR, Bruker EMX PLUS, Bruker, Karlsruhe, Germany) with a sweep width of 50 G. The electrochemical impedance and Mott–Schottky curve of the samples were tested by an electrochemical workstation (CHI660E) with the standard three-electrode testing method. The UV–Vis diffuse reflectance spectra (DRS) were obtained by a Hitachi F-7000 spectro-photometer.

### 2.3. Photocatalytic Evolution

The photocatalytic hydrogen evolution of the samples was tested through a gas chromatograph (GC-7900, Techcomp (China) Co., Ltd., Shanghai, China), which was used to measure the hydrogen production of samples under simulated sunlight irradiation. Before the experiment, 100 mg of catalyst sample was added into the mixture of 50 mL of Na_2_S (0.35 M) solution and 50 mL of Na_2_SO_3_ (0.25 M) solution, wherein Na_2_S and Na_2_SO_3_ were used as sacrificial agents in the photocatalytic hydrogen production experiment. During the experiment, a PLS-SE300C 300 W xenon lamp was used to simulate the solar light source for testing.

## 3. Results and Discussion

The qualitative phase analysis and crystal structure analysis of photocatalysts could be carried out through an x-ray diffraction test. Figure 1 shows the XRD patterns of Mn-SrTiO_3_@CF, SrTiO_3_@CF, carbon fibers, and pure SrTiO_3_. It can be observed from Figure 1 that the SrTiO_3_@CF photocatalytic composite fibers have obvious diffraction peaks at 32.25°, 39.72°, 46.38°, 57.70°, 67.61°, and 76.85°, respectively, corresponding to (110), (111), (200), (211), (220), and (310) crystal faces of SrTiO_3_ (PDF#35-0734), respectively, which indicates that the loaded SrTiO_3_ has a cubic phase perovskite structure [27,28]. Mn-SrTiO_3_@CF have basically the same diffraction peak positions and peak intensities as SrTiO_3_@CF and have no new diffraction peak, from which it can be presumed that the doping of 5% Mn does not significantly change the crystal structure of the composite catalyst [13]. The above test analysis indicated that strontium titanate semiconductor material was successfully loaded on the surface of carbon fibers, but whether Mn was doped into the strontium titanate material needed to be further tested and verified.

The morphology characteristics of Mn-SrTiO_3_@CF photocatalytic composite fibers and its preparation process can be further understood through an SEM test and characterization. As shown in Figure 2a,b, Tencel fibers are smooth-faced circular elongated fibers with a diameter of about 10 µm. Figure 2c shows the morphology structure of SrTiO_3_@Tencel fiber prepared by the solvothermal method; paste-like SrTiO_3_ particles are coated on the surface of Tencel fibers, and the diameter of SrTiO_3_@Tencel fiber composite fibers is about 11 µm. It can also be seen from Figure 2c that the SrTiO_3_ layer on the surface of Tencel fibers has obvious cracks, which may be caused by high pressure irradiation in the process of taking SEM pictures. This is because, under high pressure irradiation, the SrTiO3 material layer would split with the Tencel fiber. In this solvothermal experiment, the Tencel fibers did not dissolve in the high temperature solution because of the use of ethylene glycol and absolute ethyl alcohol as solvents, which effectively prevented cellulose fibers from dissolving. Figure 2d–f shows the morphology structure of SrTiO_3_@CF composite fibers prepared after the Tencel fibers are carbonized at a high temperature to form carbon fibers, from which it can be observed that SrTiO_3_ particles are closely coated on the surface of carbon fibers to form a coating layer with a core shell structure. SrTiO_3_ particles are organically combined with carbon fibers by the high temperature carbonization process to form a firm and close whole. Interestingly, it can be observed from Figure 2f that SrTiO_3_ nanoparticles have a cubic phase structure, which is consistent with the XRD test results of SrTiO_3_@CF. In addition, it can be found by comparing Figure 2c,d that the diameter of carbon fibers formed by carbonizing Tencel fibers at a high temperature is reduced, and the diameter of the prepared SrTiO_3_@CF composite fibers is 6–7 µm. Figure 2g–h is an SEM diagram of Mn-SrTiO_3_@CF composite fibers prepared after doping of 5% Mn, with the surface morphology structure similar to that of SrTiO_3_@CF composite fibers. Figure 2i shows an SEM diagram of the cross-section structure of Mn-SrTiO_3_@CF composite fibers, from which it can be observed that the carbon fiber in the core of the composite material has a circular structure, and the thickness of the Mn-SrTiO_3_ catalyst layer of the shell layer is about 100 nm.

To explore the distribution of various elements in the Mn-SrTiO_3_@CF sample, the SEM mapping test was performed, and the results are shown in Figure 3. It can be observed from Figure 3b that the color of carbon fibers is lighter, which is consistent with the test results in [29]. This is because the SEM mapping test is mainly for the element distribution on the surface of the samples within a certain area, the Mn-SrTiO_3_@CF composite fibers have a core shell structure, and the carbon fibers are in the coated state, so the content of C element detected in the Mn-SrTiO_3_@CF is relatively small. As shown in Figure 3, elements such as Sr, Ti, Mn, and O are evenly distributed on the surface of carbon fibers, and the content of Mn is lower, which corresponds to the actual ratio.

TEM and HRTEM of the Mn-SrTiO_3_@CF sample are shown in Figure 4. SrTiO_3_ plies observed in Figure 4a are nano-fragments shed from the Mn-SrTiO_3_@CF sample that was cut into pieces, from which it can be seen that most of the SrTiO_3_ plies are composed of small particles of 10–20 nm combined together, and these small nanoparticles have a larger specific surface area, which is conducive to improving the photocatalytic activity. The corresponding interplanar spacing shown in Figure 4c is about 0.279 nm, which shall belong to the (110) crystal face of SrTiO_3_ (PDF#35-0734) [30]. The doping of a small amount of Mn may change the spacing of the crystal face, which is not significant. The above experimental results showed that Mn was successfully doped in the SrTiO_3_ material [31], and the doping of a small amount of Mn does not significantly change the lattice structure of the SrTiO_3_ material.

The composition and valence state of Mn-SrTiO_3_@CF photocatalysts are researched through X-ray photoelectron spectroscopy (XPS). Figure 5 shows the full spectrum of the Mn-SrTiO_3_@CF sample and the high-resolution XPS spectra of C 1s, Sr 3d, Ti 2p, O 1s, and Mn 2p. Figure 5a shows the full spectrum of the Mn-SrTiO_3_@CF sample, indicating that there are mainly Sr, Ti, Mn, O, and C in the prepared sample, without other impurities. In the fitting spectrum of C 1s shown in Figure 5b, the fitting peak at 284.0 eV is a C–C bond, which corresponds to the C–C bond in carbon fibers and is partially similar to the graphite structure, while the fitting peak at 285.2 eV shall belong to a small amount of C–O bonds on the surface of carbon fibers [31]. It is presumed that carbon fibers and SrTiO_3_ material are bonded partially through a chemical bond formed by O atoms. It can be observed from Figure 5c that the high resolution XPS spectrum of Ti 2p can be fitted into two characteristic peaks, which are located at the binding energies of 458.0 eV and 463.8 eV, respectively, which, presumably, shall belong to Ti 2p_3/2_ and Ti 2p_1/2_, respectively, corresponding to Ti^4+^ ions. The difference between the two fitting peaks is 5.8 eV [8]. It can be observed from the spectrum of Sr 3d shown in Figure 5d that peaks at the binding energies of 132.5 eV and 134.3 eV are attributed to Sr 3d_5/2_ and Sr 3d_3/2_ of Sr^2+^, and the difference between the peaks is 1.8 eV [32]. Interestingly, as shown in Figure 5e, the peak of O 1s is fitted into three peaks that are located at 529.2 eV, 530.9 eV, and 532.4 eV, respectively, wherein the peak at the binding energy of 529.2 eV corresponds to lattice oxygen in SrTiO_3_, while the peak at the binding energy of 530.9 eV may be adsorption oxygen in oxygen vacancy, and the peak at 532.4 eV corresponds to surface adsorption oxygen in the SrTiO_3_ catalyst. This test indicated that there may be oxygen vacancies in the Mn-SrTiO_3_@CF catalyst sample [33,34], which needs to be further confirmed by the EPR test. Figure 5f shows the high-resolution XPS diagram of Mn 2p. Because of the low content of doped Mn, the corresponding measured XPS characteristic peak signal is relatively weak. After peak fitting, it is presumed that the characteristic peaks located at the binding energies of 641.6 eV and 652.6 eV correspond to Mn 2p_3/2_ and Mn 2p_1/2_, respectively, and the difference between the two peaks is 11 eV.

The EPR test was conducted to further confirm the existence of oxygen vacancies in the Mn-SrTiO_3_@CF sample, as shown in Figure 6. It can be observed from Figure 6 that the Mn-SrTiO_3_@CF sample has a strong peak at g ≈ 2.003, indicating the existence of oxygen vacancies in the Mn-SrTiO_3_@CF sample [34,35]. The generation of oxygen vacancies in the Mn-SrTiO_3_@CF sample may be caused by the fact that, in the carbonization process of bamboo pulp fibers loaded with SrTiO_3_, a part of C reacts with O in SrTiO_3_, and after SrTiO_3_ is treated at high temperature in argon atmosphere, a part of O in the lattices sheds off, thus forming oxygen defects. The existence of these oxygen vacancies will help to expand the light absorption range of the photocatalyst as well as to improve the charge transfer ability of the Mn-SrTiO_3_@CF material. Combined with the above analysis of results, it indicated that Mn-SrTiO_3_@CF composite material rich in oxygen vacancies is successfully prepared, and the doping of Mn will further improve the photocatalytic activity of the material.

The photocatalytic hydrogen production performance of composite materials such as SrTiO_3_@CF and Mn-SrTiO_3_@CF was tested under simulated sunlight. The test results are shown in Figure 7. Na_2_S and Na_2_SO_3_ were used as sacrificial agents in this experiment. It can be seen from Figure 7 that carbon fibers have no hydrogen production performance under the action of light, while the hydrogen production performance of SrTiO_3_@CF photocatalytic composite fibers is about 46.90 μmol/g·h. It is presumed that the existence of oxygen vacancies in the SrTiO_3_@CF material promotes the extension of its light absorption boundary to visible light and enhances the charge transfer ability, which is conducive to improving the photocatalytic property of the material. Meanwhile, the carbon fibers have functions similar to co-catalysts, promoting the migration of photoelectrons [24]. As shown in Figure 7, the hydrogen production capacity of Mn-SrTiO_3_@CF composite photocatalytic fibers reaches 285.37 μmol/g·h, which is about six times that of the SrTiO_3_@CF material, which is obviously attributed to the doping of a small amount of Mn ions; the result is similar to the research in [36]. The doping of Mn ions not only promoted the red shift of the light absorption boundary and the extension to visible light, but also improved the separation and migration efficiency of photocarriers. The above test results showed that the existence of oxygen vacancies, the function of carbon fibers similar to a co-catalyst, and the doping of Mn ions significantly improved the hydrogen production performance of Mn-SrTiO_3_@CF photocatalytic composite fibers.

The cyclic stability of the Mn-SrTiO_3_@CF composite catalyst was tested, and the results are shown in Figure 8. After four consecutive tests of cyclic photocatalytic water splitting hydrogen production performance, the average photocatalytic hydrogen production performance of the Mn-SrTiO_3_@CF composite catalyst is about 267.69 μmol/g·h. There was only a slight decrease over a period of cyclic experiments, indicating that the the Mn-SrTiO3@CF composite catalyst can maintain relatively stable photocatalytic performance in the water splitting hydrogen production reaction.

The light absorption of catalyst materials such as Mn-SrTiO_3_@CF can be measured by UV–Vis diffuse reflection spectrum test. As shown in Figure 9a, pure SrTiO_3_ nanoparticles have an obvious characteristic absorption edge at 375 nm, while the light absorption property of SrTiO_3_@CF is significantly enhanced compared with pure SrTiO_3_, which mainly comes from the strong light absorption property of carbon fibers. Mn-SrTiO_3_@CF photocatalytic composite fibers have stronger light absorption property compared with SrTiO_3_@CF materials, because the doping of Mn reduces the band gap of the composite material and promotes the red shift of the light absorption boundary and the extension to visible light [37]. Mn-SrTiO_3_@CF and SrTiO_3_@CF materials have a significant change in the radian of the bottom of the light absorption edge, which is presumed to be because of the existence of oxygen vacancies in the material, reducing the band gap of the catalyst and further enhancing the light absorption property.

Based on the Kubelka–Munk theory, the band gap (*E*_g_) of semiconductor materials can be calculated according to Equation (1) [38]. As shown in Figure 9b, *hν* is drawn with (A*hν*)^2^, from which it can be obtained that the band gap of single SrTiO_3_ is about 3.32 eV. Similarly, it can also be calculated that the band gaps of Mn-SrTiO_3_@CF and SrTiO_3_@CF composites are about 2.91 eV and 3.08 eV, respectively.
(1)$$\left(Ahv{)}^{2}=C(hv-E_{g}\right)$$
where *A* is absorbance in UV–visible diffuse reflection; *hν* is photon energy, which is replaced here by the 1024/ wavelength; and *C* is a constant.

The band structure is an important factor affecting the photocatalytic property, and the flat band potential of the semiconductor catalyst can be calculated using the Mott–Schottky equation (Equation (2)) [38,39].
(2)$$\frac{1}{C^{2}}=\frac{2}{\varepsilon \varepsilon_{0}eN_{D}}(V-V_{FB}-\frac{k_{B}T}{e})$$
where *C* is the interface capacitance and *V_FB_* is the flat band potential.

As shown in Figure 10, the tangent slope of the Mott–Schottky spectral line of SrTiO_3_ is positive, indicating that SrTiO_3_ is an n-type semiconductor material and the flat band potential of SrTiO_3_ is −0.69 eV (calomel electrode, vs. SCE). Based on the fact that the potential of a calomel electrode relative to a standard hydrogen electrode at 25 °C is about 0.24 eV, it can be calculated that the Fermi level corresponding to SrTiO_3_ is about −0.45 eV. It is generally believed that the conduction band position of n-type semiconductor is 0.1 eV different from the Fermi level [40], so the conduction band position of SrTiO_3_ is −0.55 eV. It can be known from Figure 9b that the band gap of SrTiO_3_ is 3.32 eV. It can be calculated that the valence band position of SrTiO_3_ is 2.77 eV.

Based on the above research results, we proposed the photocatalytic water splitting hydrogen production mechanism of the Mn-SrTiO_3_@CF photocatalytic composite material, as shown in Figure 11.

According to the test results of the UV–Vis diffuse reflection spectrum (Figure 9), it can be calculated that the composite band gaps of the pure SrTiO_3_, Mn-SrTiO_3_@CF, and SrTiO_3_@CF composite materials are about 3.32 eV, 2.91 eV, and 3.08 eV, respectively. The results indicated that the doping of Mn and the existence of oxygen vacancies reduced the band gap of the corresponding material; expanded the light absorption range, which extended to the visible light; and improved the photocatalytic activity. The Mn-SrTiO_3_@CF composite material has a large number of oxygen vacancies, which will create a new donor level below the conduction band, constituting an oxygen vacancy state (VOs) and becoming the capture center of photoelectrons. As shown in Figure 11, under the action of light, the Mn-SrTiO_3_ composite material produces electrons and holes, and photoelectrons migrate to the conduction band and the oxygen vacancy state (VOs) [41]. Some of the electrons located on the conduction band will migrate to the surface of carbon fibers, and some will migrate to the oxygen vacancy state (VOs), which promotes the separation and migration of photoelectrons and holes, thus improving the photocatalytic property. The electrons on the conduction band and the oxygen vacancy state (VOs) will combine with H^+^ ions in water to produce hydrogen, while the holes on the valence band will combine with sacrificial agents (Na_2_S and Na_2_SO_3_) in aqueous solution to promote the separation and generation of photoelectron–hole pairs.

## 4. Conclusions

In this work, Tencel fibers were taken as the substrate, and SrTiO_3_@CF and Mn-SrTiO_3_@CF with a firm structure were successfully obtained through the process route of first loading the semiconductor material on the carrier and then carbonizing the tencel fibers. This solved the problem that the semiconductor materials were difficult to directly load on the surfaces of carbon fibers or easy to shed off because of the smooth surface and few active groups of carbon fibers. The Mn-SrTiO_3_@CF composite photocatalytic fibers exhibited a higher activity for hydrogen evolution compared with the SrTiO_3_@CF material. Particularly, the photocatalytic hydrogen production of the Mn-SrTiO_3_@CF composite catalyst is about 267.69 μmol/g·h with 5% Mn-doped, which is six times that of the SrTiO_3_@CF material. After modifying SrTiO_3_@CF with Mn, the light absorption boundary could be extended to the visible light direction, and the separation and migration efficiency of photocarriers could be improved. In addition, SrTiO_3_@CF and Mn-SrTiO_3_@CF photocatalytic materials rich in oxygen vacancies were successfully prepared through the high temperature carbonization process. The existence of oxygen vacancies would generate a new donor level below the conduction band, constituting an oxygen vacancy state (VOs) and becoming the capture center of photoelectrons, thus significantly improving the photocatalytic activity. The synergistic effect from Mn doping, oxygen vacancies, the sacrificial agent, and carbon fibers can efficiently absorb photons, transfer photoinduced electrons, restrain carrier recombination, and improve the efficiency of the catalyst hydrogen production.

## Figures and Tables

**Figure 1 materials-15-04723-f001:**
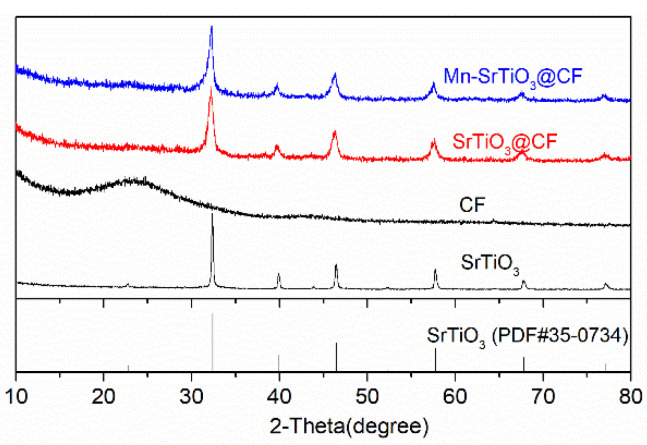
XRD pattern of samples.

**Figure 2 materials-15-04723-f002:**
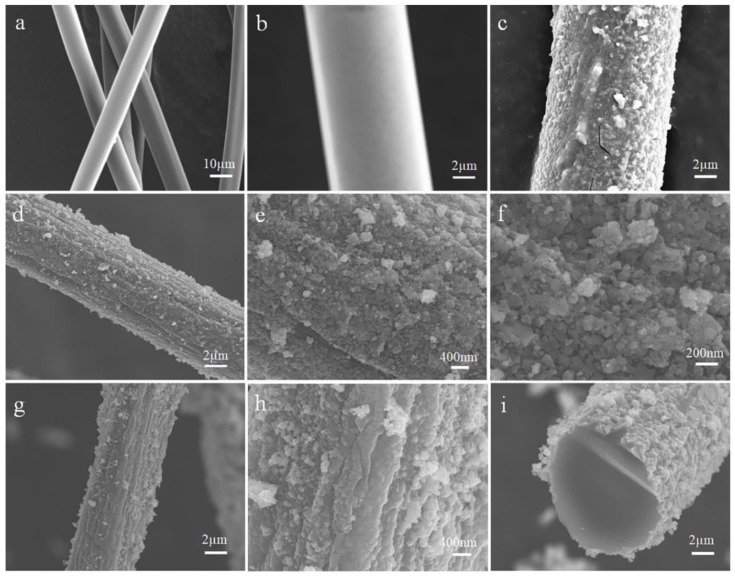
SEM diagram of samples: (**a**–**c**) Tencel fiber, (**c**) SrTiO_3_@Tencel fiber, (**d**–**f**) SrTiO_3_@CF, (**g**–**i**) Mn-SrTiO_3_@CF.

**Figure 3 materials-15-04723-f003:**
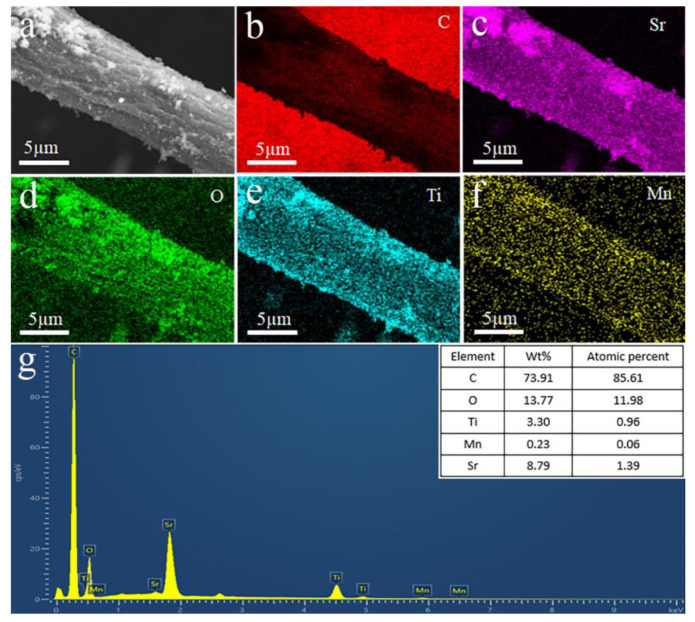
SEM mapping diagram of Mn-SrTiO_3_@CF sample: (**a**) SEM diagram of composite fiber, (**b**) C element, (**c**) Sr element, (**d**) O element, (**e**) Ti element, (**f**) Mn element, and (**g**) element content diagram.

**Figure 4 materials-15-04723-f004:**
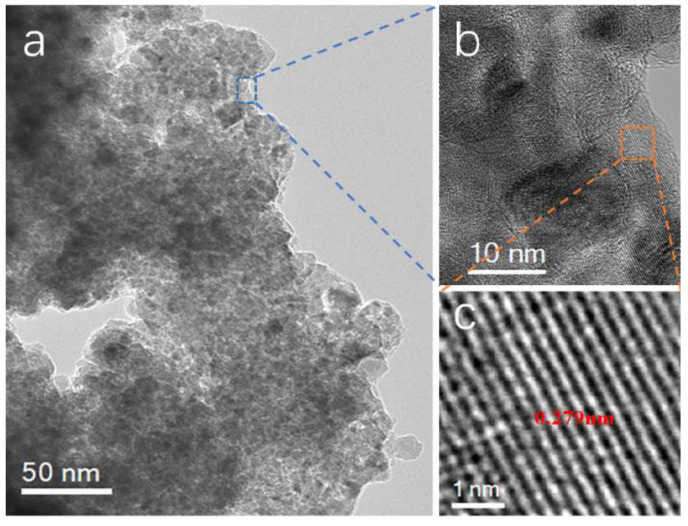
TEM diagram of the Mn-SrTiO_3_@CF sample, (**a**) Mn-SrTiO_3_ nanosheets, (**b**) HRTEM at the interface, and (**c**) HRTEM of Mn-SrTiO_3_.

**Figure 5 materials-15-04723-f005:**
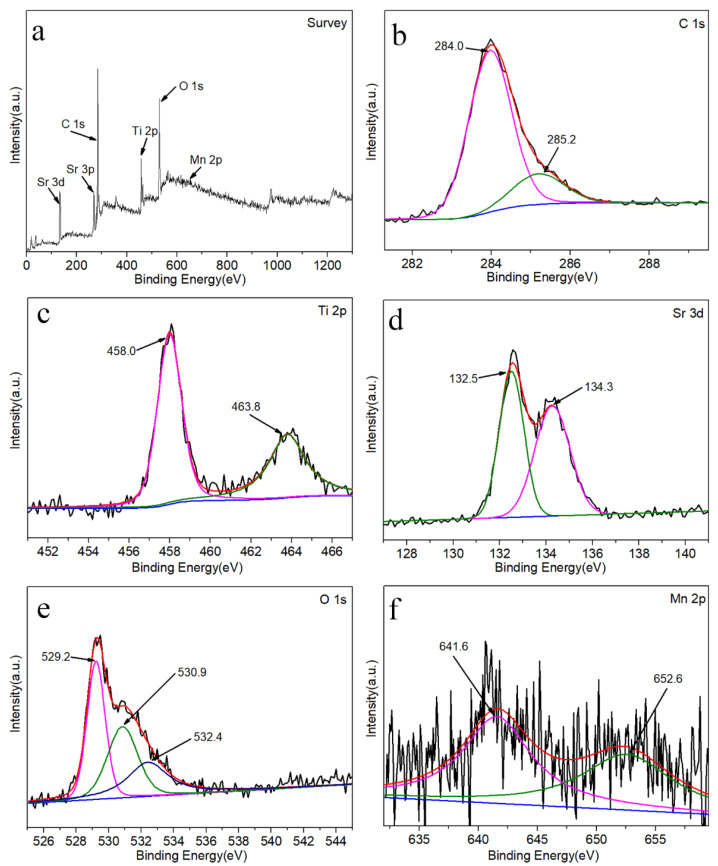
XPS diagram of Mn-SrTiO_3_@CF sample, (**a**) full spectrum, (**b**) C 1S spectrum, (**c**) Ti 2P spectrum, (**d**) Sr 3D spectrum, (**e**) O 1S spectrum, and (**f**) Mn 2P spectrum.

**Figure 6 materials-15-04723-f006:**
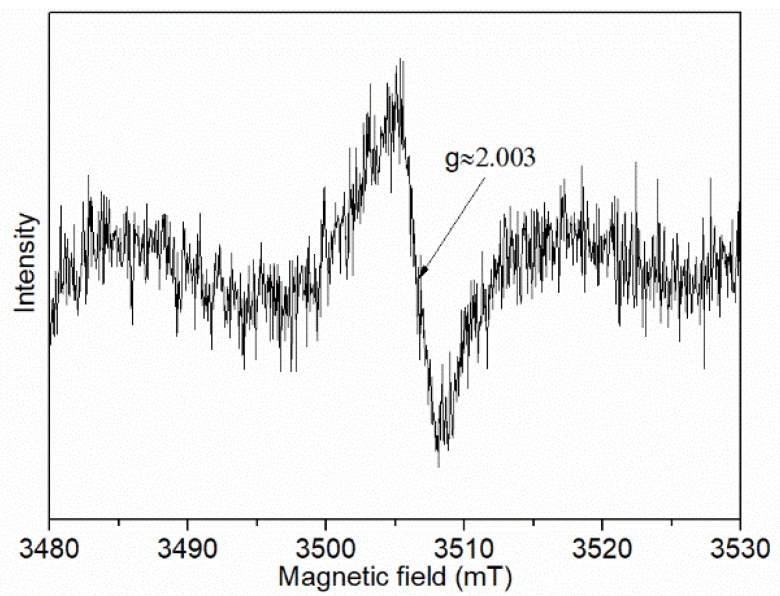
EPR test of the Mn-SrTiO_3_@CF sample.

**Figure 7 materials-15-04723-f007:**
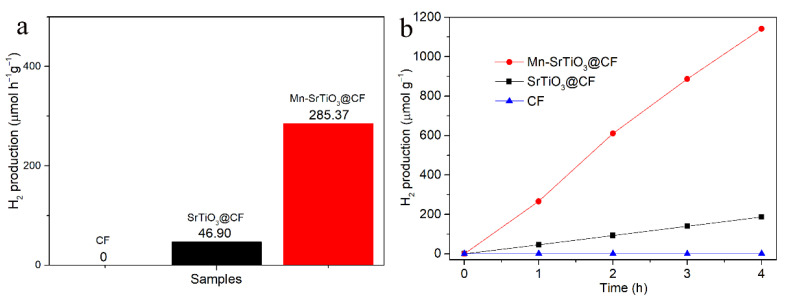
Photocatalytic hydrogen production rate (**a**) and time-dependent hydrogen production amount of different samples (**b**).

**Figure 8 materials-15-04723-f008:**
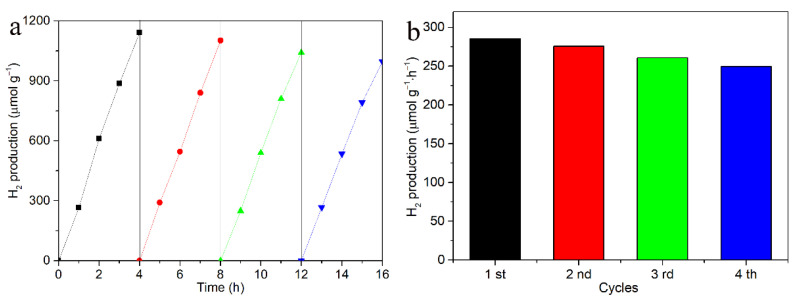
Photocatalytic stability diagram of Mn-SrTiO_3_@CF composite material. (**a**) Cycle curve of hydrogen production amount, (**b**) The corrresponding histogram.

**Figure 9 materials-15-04723-f009:**
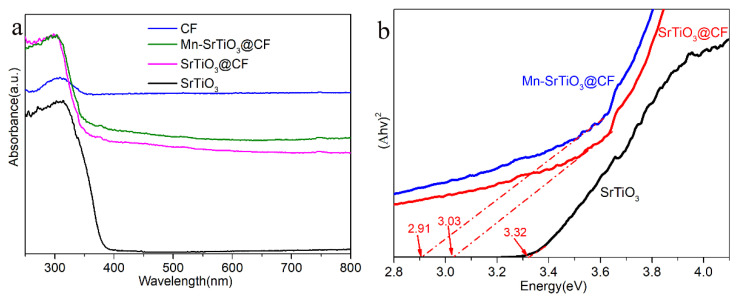
UV–Vis diffuse reflection spectra of samples (**a**) and band gap of the SrTiO_3_ sample (**b**).

**Figure 10 materials-15-04723-f010:**
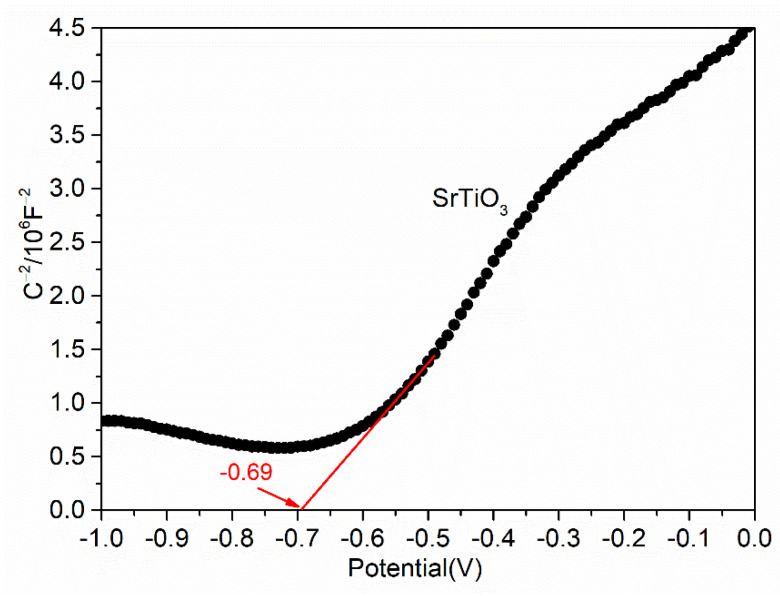
Mott–Schottky plot of the SrTiO_3_ sample.

**Figure 11 materials-15-04723-f011:**
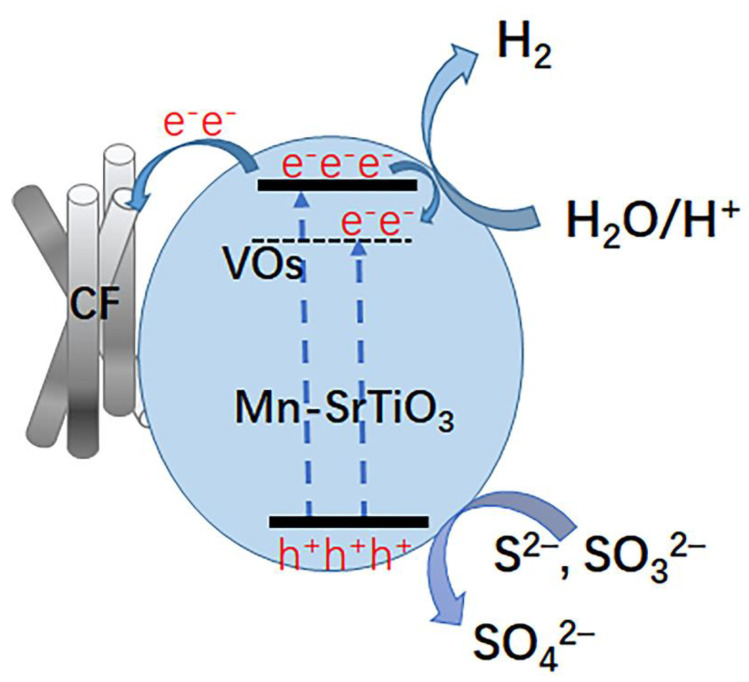
Hydrogen production mechanism of Mn-SrTiO_3_@CF photocatalytic composite material.

## Data Availability

Data is contained within the article.

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
