# Peer review of "Fabrication of Mn-Doped SrTiO3/Carbon Fiber with Oxygen Vacancy for Enhanced Photocatalytic Hydrogen Evolution"

_materials, 2022, doi:10.3390/ma15134723_

Round 1
Reviewer 2 Report
This manuscript presented the preparation of Mn-doped SrTiO3/Carbon Fiber and its application for photocatalytic hydrogen evolution. While this topic is very interesting, this manuscript needs a major revision before it could be accepted.
1. Language needs to be further polished.
2. The authors claimed that the peak at the binding energy of 530.9 eV may be adsorption oxygen in oxygen vacancy, and the peak at 532.4 eV corresponds to surface adsorption oxygen in the SrTiO3 catalyst. Do you have any literature to support this claim?
3. How would your materials compare with the state-of-the-art photocatalytic hydrogen evaluation catalysts in terms of hydrogen production capacity? I would recommend the authors to have a table to compare the hydrogen production capacity of their work to different catalysts materials from literature, so that the general audience could have a better understanding how significant this work is.
4. The authors discussed a lot about the “controllable introduction” of oxygen vacancy in the Introduction part. However, I didn’t see any discussions or experiments that talked about how their method could control the introduction of oxygen vacancies. Besides, their comments on the generation of oxygen vacancy using their preparation method are not very convincing. Moreover, what is the role of Mn doping, and what is the role of oxygen vacancy exactly?
Reviewer 3 Report
The article is devoted to the study of the properties of SrTiO3 nanoparticles deposited on Tencel fibers by the solvothermal method. In general, the presented study is quite interesting and promising not only from a fundamental point of view, but also from a further practical application. The article corresponds to the declared journal and can be accepted for publication in the future after the authors answer a number of questions that have arisen during its analysis.
1. The abstract needs to be improved, the authors should reflect in more detail the novelty and practical significance of the work.
2. The authors should present the results of phase analysis, taking into account the possibility of assessing the degree of crystallinity and the amorphous phase of the samples.
3. The authors should explain how the efficiency of photocatalytic activity increased with the addition of Mn?
4. The authors should describe in more detail how exactly the number of oxygen vacancies in the structure of the material was determined, and how they affect the properties of the material.
5. You should also pay attention to the catalytic properties of the obtained structures, how much do they differ from commercial analogues and how effective are they in application compared to other types of materials?
6. Conclusion requires significant revision and optimization, as well as reflection of further research prospects.
Reviewer 4 Report
This manuscript reports an experimental study on the fabrication of M-doped SrTiO3 loaded carbon fiber from Tencel fibers. Cabonization after solvothermal synsthesis of oxide is effective and useful for the readers. However, there are many points that must be amended before further consideration.
1. Many misspelling: For example, "sloid" in abstract maybe "solid". There are many others. It is author's responsibility to check them.
2. Incorrect wording. For example, page 3. "XRD energy spectrum" should be "XRD (powder) pattern". The x-axis of Fig. 1 is not energy but angle.
3. Durability of phohotocatalyst: Fig.8 shows clear decrease over repeating time. It must be mentioned in the text.
4. page 12: The following claim must be supported by citing reference. p12 "It is generally believed that the conduction band position of n-type semiconductor is 0.1 eV different from the Fermi level,"
5. Figs. 9-11: Kubelca-Munk and Mott-Schottoky analysis of all samples must be shown for in the figures.
6. page 7 bottom: Mn XPS peak energies and separation (11eV) are different from Ref. 13. Doesn't this mean Mn states of this sample is different from that in Ref. 13 and cannot be compared directly? It may be from the different solvothermal conditions.
Round 2
Reviewer 2 Report
This paper was improved but I still believe that the general audience will expect a comparison of this work with other catalyst materials in terms of performance.
Reviewer 3 Report
The authors answered all the questions, the article can be accepted for publication.